# The Mediating Role of Vision in the Relationship between Proprioception and Postural Control in Older Adults, as Compared to Teenagers and Younger and Middle-Aged Adults

**DOI:** 10.3390/healthcare10010103

**Published:** 2022-01-05

**Authors:** Ainhoa Nieto-Guisado, Monica Solana-Tramunt, Adrià Marco-Ahulló, Marta Sevilla-Sánchez, Cristina Cabrejas, Josep Campos-Rius, Jose Morales

**Affiliations:** 1Department of Sports Sciences, FPCEE Blanquerna, Ramon Llull University, 08022 Barcelona, Spain; ainhoang@blanquerna.url.edu (A.N.-G.); cristinacm19@blanquerna.url.edu (C.C.); josepcr@blanquerna.url.edu (J.C.-R.); josema@blanquerna.url.edu (J.M.); 2Department of Neuropsychology, Methodology, Social and Psychology, Faculty of Psychology, Catholic University of Valencia, 46001 Valencia, Spain; adria.marco@vhir.org; 3Faculty of Sports Sciences and the E.F., University of Coruña, 15179 Coruña, Spain; marta.sevilla@udc.es

**Keywords:** motor control, balance, repositioning error, injury prevention

## Abstract

The aim of this study is to analyze the mediating role of vision in the relationship between conscious lower limb proprioception (dominant knee) and bipedal postural control (with eyes open and closed) in older adults, as compared with teenagers, younger adults and middle-aged adults. Methods: The sample consisted of 119 healthy, physically active participants. Postural control was assessed using the bipedal Romberg test with participants’ eyes open and closed on a force platform. Proprioception was measured through the ability to reposition the knee at 45°, measured with the Goniometer Pro application’s goniometer. Results: The results showed an indirect relationship between proprioception and postural control with closed eyes in all age groups; however, vision did not mediate this relationship. Conclusions: Older adults outperformed only teenagers on the balance test. The group of older adults was the only one that did not display differences with regard to certain variables when the test was done with open or closed eyes. It seems that age does not influence performance on proprioception tests. These findings help us to optimize the design of training programs for older adults and suggest that physical exercise is a protective factor against age-related decline.

## 1. Introduction

Poor postural control is a common problem among older adults, and it is associated with mobility issues, limitations on everyday activities and the risk of falling [1]. Postural control is defined as the ability to maintain the body’s center of gravity within the base of support [2]. This motor skill is closely connected to the central nervous system, and it evolves with age [3], reaching complete maturity in adulthood (>18 years of age). This maturity is followed by a decline, when people gradually lose synapses and their non-functional neural connections disappear, causing a deterioration of motor skills among older adults [4]. One way this manifests itself is that older adults tend to display greater variability in the center of pressure than younger adults when carrying out postural tasks [5]. Postural control depends on the interaction of proprioceptive receptors, the sense of touch, the vestibular system and the processing of visual information, all of which helps regulate tone and the perception of force and pressure [6]. The degree to which this skill reaches its highest potential depends on the integration of visual, vestibular, proprioceptive and tactile inputs, which play a role in regulating the tonicity and the perception of strength and pressure needed to keep the balance [6].

The aging process seems to involve a deterioration in proprioceptive functioning [7] and, consequently, a loss of postural control [8]. Proprioception is a facet of kinesthetic perception of the perceptual motor qualities. It was originally defined as the functional information we gather from our joints and muscles that allows us to be aware of our bodies’ movements and positioning or to respond unconsciously to involuntary changes in joint position, all so as to maintain balance as well as muscle tone and coordination [9]. Vision also plays a key role in postural control. A lack of visual information affects both proprioception and postural control (see the review article by [7]). For example, researchers have found that both younger and older adults display greater postural oscillation when their eyes are closed then when they are open [10]. When we make certain body movements, our sense of sight is often occupied by processing information not directly connected to the movement itself, meaning that the central nervous system has to rely on other stimuli such as proprioception as it ensures the body’s postural control [11,12]. Thus, in the absence of vision, the nervous system facilitates the entry of information from other sensory stimuli (such as proprioception) into the primary somatosensory cortex of the parietal lobe [13]. Additionally, when we focus our attention on the stability of a joint, the central nervous system prioritizes proprioceptive stimuli in order to boost our awareness of our joint positioning, thus allowing for more effective postural control associated with the joint [14,15]. The cause of this deterioration of postural control is unclear, but conscious proprioception, vision and the vestibular system all play important roles [8]. Nonetheless, when visual information is compromised, there does not seem to be any effect on the worsening of postural control associated with aging [16,17]. Therefore, it is likely that the decline in postural control among elderly people has its origin in the deterioration of the peripheral nervous system and in the decrease in the brain’s central processing capacity, both of which affect proprioceptive acuity and, consequently, postural control [17].

Currently, training and stimulation of proprioception and postural control are important elements of readaptation and rehabilitation regimens [18], as there is a consensus that these abilities are key to the prevention of lower limb injuries [19]. This is especially true of knee injuries, as this joint, along with the hips, ankles and toes, bears the greatest responsibility for balance [20]. According to [21], a number of studies have featured exercise sessions involving imbalance and other efforts to increase postural control, all with an eye toward improving proprioception and reducing the likelihood of injury. However, recently researchers have found that not all proprioception programs are effective [22]. It is widely accepted that in order to train any quality of the human body it is necessary to subject the systems and organs involved to a level of stress that goes beyond their immediate capacities. The body responds to this load with a positive adaptation in the quality targeted for improvement [23]. Different mechanoreceptors are involved in proprioception and postural control, meaning that when stimulating the receptors that affect the latter, one is not necessarily stimulating the ones involved in the former. Thus, not all postural control exercises enhance proprioception [22,24]. 

Although it is well known that proprioception plays a fundamental role in postural control, there unfortunately has, to our knowledge, been no research into the mediating role that vision might play between conscious proprioception and postural control. Greater knowledge of how postural control works and how it evolves throughout our lives could help professionals design better training, readaptation and rehabilitation programs. Assessment via static proprioception tests (such as joint positioning perception) and postural control tests (such as measurements of bipedal posture on a force plate) could be useful tools to help predict the risk of falls among older adults (see the review article by [25]). These tests are also easy to carry out and more accurate than dynamic tests [16], and they can provide us with valuable information about people’s general sensory motor performance [26]. 

In light of the above, the main goal of this study will be to analyze the mediating role of vision in the relationship between conscious lower limb proprioception (dominant knee) and bipedal postural control (with eyes open and closed) in older adults, as compared with teenagers, younger adults and middle-aged adults. The research hypothesis is that there will be no difference between the age groups in terms of the mediating role played by vision, as the decline in conscious proprioceptive performance that comes with age, and the resulting decrease in postural control, seem to have their origins in the deterioration of the nervous system.

## 2. Materials and Methods

### 2.1. Design

The study used a randomized repeated measures cross-sectional design in order to investigate any differences between the participants’ proprioceptive performance and postural control test results in terms of their age.

### 2.2. Participants

The overall sample for this study was made up of 119 physically active individuals [according to the results of the International Physical Activity Questionnaire (IPAQ) 4611.03 ± 1620.97; Metabolic Equivalents (METS)] of between 12 and 85 years of age. They were recruited from a number of different educational centers and sports facilities. The sample was divided into the following age groups: teenagers (12–18); young adults (19–35); middle-aged adults (36–64); and older adults (65–85) (Table 1). Potential participants were excluded from the study if they: (i) had been diagnosed with a neurological pathology such as Alzheimer’s disease or Parkinson’s disease, (ii) were unable to carry out all the parts of the test and/or (iii) were undergoing pharmacological treatments that could alter their normal functioning in the tests iv) or had had joint replacement surgery on a lower limb. 

Prior to data collection, the software program G * POWER (Düsseldorf FRG, Department of Psychology of the University of Düsseldorf, Düsseldorf, Germany) was used to calculate the sample size required to obtain a Power (1 − ß) > 0.9 and an effect size of 0.35. The level of significance was set at *p* < 0.05 in all analyses. All the statistical analyses were carried out using the Statistical Package for Social Science version 24.0 software (SPSS, Inc., Chicago, IL, USA).

The study was carried out in accordance with the ethical standards set out in the Helsinki Declaration, and it was approved by the ethics committee of Universitat Ramon Llull, ID number 1718006D. All the participants in the study were informed of the procedure and signed informed consent documents.

### 2.3. Procedure

A few days before carrying out the measurement protocol, the researchers visited the institutions from which the participants had been recruited, in order to collect data by an anonymous formular sent by email to the participant. Prior to sending the email we check that older adults were familiar with the internet, and we gave them verbal instructions to fulfilling the formular. This included information on the participants’ physical characteristics, along with data to determine whether the individuals fulfilled the inclusion/exclusion criteria, not medical history was collected. In accordance with con [27] the participants’ lower limb dominance was established through a self-report on their performance on bilateral movement tasks involving the legs. In order to ensure that the order of the tests did not influence the results, the instruments were administered in a randomized order to each of the participants. Finally, each participant completed the short version IPAQ [28,29] to provide us with information about his or her self-reported degree of physical activity.

### 2.4. Proprioception

In order to assess proprioception, we calculated the absolute error (AEr) when participants attempted to reposition their dominant knee at a 45º angle. This was done using a valid and reliable mobile application called Goniometer Pro (2.9, FiveFufFive co, Bloomfield, NJ, United States), which was installed on a Galaxy J7 (Samsung, Seoul, South Korea) smartphone. With this application, which was validated in a prior study [30], it is possible to use your mobile phone as a digital goniometer. It provides instantaneous, accurate, repeatable readings (α de Cronbach de 0.993 and ICC 0.993) of the range of movement (ROM), measuring the difference between the required knee angle and the position that is actually adopted [31].

First, the participant was outfitted with orthopedic boots (Figure 1) on both legs in order to cancel out the effects of any balance adaptation made by the toes or the ankles, allowing us to focus exclusively on the knees. Then, the smartphone was placed so as to be aligned with the longitude of the femur, with the base of the device aligned with the femur-tibia joint space.

While previous studies have made use of ski boots to limit the range of toe and ankle movement [32], the researchers here opted for the use of orthopedic boots because they are easier to put on and remove. It was also possible to use these boots with all the participants regardless of the size of their feet. The boots are fastened with adjustable cords that adapt to any size of foot and restrict plantar flexion and dorsiflexion.

Once the participants had been outfitted with these instruments, the researchers asked them to close their eyes. The researchers placed each participant’s knee joint at a 45º angle and asked them to maintain this position for six seconds. Then, the participants were asked to return to the original standing position with both feet on the floor. After that, when they were ready, they were told to bend their knees again until they had returned to exactly the same position the researcher had placed them in when their eyes had been closed. Upon repositioning, the angle was measured three times, with the final figure taken from the mean of the three measurements. The absolute value of the difference between the requested (45°) and the actual angle was recorded to determine the participant’s joint repositioning ability, the variable used to quantify conscious proprioception [31].

### 2.5. Postural Control

Postural control was measured using a Kistler force plate (Kistler Instruments AG, Winterthur, Switzerland) connected to a laptop computer running the Kistler MARS 3.0 software program. In order to assess the participants’ bipedal postural control, the researchers asked them to stand on the plate and complete the Romberg test under two different sets of conditions: Romberg bipedal open eyes (ROE); Romberg bipedal closed eyes (RCE). Prior to the test, the participants were given some guidelines as to how to stand up straight with their gaze fixed on a certain point two meters above their eye lines and their arms extended along their torsos. They were also asked to place their feet about shoulder-width apart. Just as they did for the proprioceptive test, the participants wore orthopedic boots on their feet, helping to guarantee that the two tests would yield comparable results. These boots served to eliminate the effects of any initial balance adaptations made by the toes or ankle, thus ensuring that the knee, along with the hips, would act as the main balancing mechanism. Each test lasted 30 s and was repeated three times under both sets of conditions, and participants rested for a minute between the attempts.

Postural control measurements were taken via center of pressure readings registered on the force plate. The values recorded were the total area and the mean velocity in the anterior–posterior (MV_AP_) and medial–lateral (MV_ML_) directions. While the total area is an indicator of the overall performance on the postural control task, the MV_AP_ and MV_ML_ readings provide information on the neuromuscular activity used to maintain postural control [33]. For all of these variables, lower scores indicate better performance on the tasks.

Data both for the variables related to proprioception and those related to postural control were gathered in a quiet room, free of distractions and interruptions. The order in which the tests were administered was chosen randomly for each participant, and the tests were repeated three times under each set of testing circumstances, with each test taking thirty seconds. Learning patterns were avoided by the researchers by keeping in silence and not giving any feedbacks across the three attempts.

### 2.6. Statistical Analysis

First, the Kolmogorov–Smirnov test was used to examine the normality of the sample, and then we carried out a calculation to determine whether there were multiple differences in the independent variables as a function of various factors.

The dependent postural control variables were examined individually via a unidirectional multivariate analysis of variance (one-way MANOVA) in order to determine if they displayed combined differences as a function of intra-subject conditions (×2: OE and CE) or in terms of the inter-subject factor of the group they belonged to (×4: teenagers, young adults, adults or older adults). A one-way ANOVA was applied to the variable proprioception to test the differences between age groups. In order to determine statistical significance, the relevant post-hoc pairwise comparisons were carried out using the Bonferroni correction.

The multivariate contrasts in each case were calculate via univariate contrast in order to determine which dependent variables had been influenced by the independent variables. Partial eta squared (η_p_^2^) was used as the effect size of both the multivariate and univariate contrasts. When the univariate contrasts showed statistically significant main effects or interaction effects, pairwise comparisons were carried out using the Bonferroni correction.

Pearson’s coefficient of correlation was used to study the relationships between the various parameters of postural control and proprioception. The relationships between the variables were assessed according to the following thresholds: trivial <0.1, small from 0.1 to 0.3, moderate from 0.3 to 0.5, large from 0.5 to 0.7, very large from 0.7 to 0.9 and nearly perfect from 0.9 to 1.0 [34].

Version 21 of the SPSS software program (SPSS Inc., Chicago, IL, EEUU) was used for all the statistical calculations, and a significance value of *p* < 0.05 was established for all statistical analyses.

Using the analysis of the correlations among the different variables, values were calculated for a new variable based on the variation between the postural control results with open and closed eyes. Then, a simple mediation analysis model was used to examine the direct and indirect effects of proprioception as a predictor variable (PV) on postural control with closed eyes as a dependent variable (DV), through the mediation of the variation of the results with eyes closed with respect to eyes open (VAR_CE, a mediating variable) (MV). The associations established through this mediation analysis are shown in Figure 2.

A simple mediation analysis indicates a total effect (c) when the PV exerts an influence on the DV without taking into account the analysis of the participation of the MV. Meanwhile, the model indicates an indirect effect (a and b) when the PV exerts its influence on the DV through the MV. Finally, a direct effect is registered when the PV has an influence on the DV in the presence of the MV, but not via this mediating variable.

The mediating effect is considered total if the PV exerts all of its influence through the MV, and it is partial if a only part of the influence comes via the MV and the other part is the result of the PV acting directly on the DV, not through the MV. In such a case, both direct and indirect influence is exerted on the DV. Therefore, both the a*b route and the c’ route are significant.

## 3. Results

The multivariate analysis carried out for all the postural control variables indicated a significant interaction effect of the group condition in MV_AP_ (F_2,114_ = 661.68; *p* < 0.001; η_p_^2^ = 0.92), MV_ML_ (F_2,114_ = 397.89; *p* < 0.001; η_p_^2^ = 0.87) and total area (F_2,114_ = 118.45; *p* < 0.001; η_p_^2^ = 0.67). There was also a significant main effect as a function of the group on the three variables MV_AP_ (F_3,115_ = 10.95; *p* < 0.001; η_p_^2^ = 0.19), MV_ML_ (F_3,115_ = 14.42; *p* < 0.001; η_p_^2^ = 0.27) and total area (F_3,115_ = 9.38; *p* < 0.001; η_p_^2^ = 0.19). The pairwise comparisons are shown in Figure 3 and Figure 4, along with the descriptive data on the postural control variables in the three blocks.

The one-way ANOVA applied to the variable of proprioception yielded no significant differences between the groups (F_3,115_ = 1.63; *p* = 0.39). Figure 5 shows the descriptive and comparative data for the groups for proprioception.

The dots represent the individual scores for absolute error, and the horizontal lines represent the mean score for each group. Lower scores reflect a better performance on the task. O: Older adults; MA: middle-aged adults YA: young adults; T: teenagers. * *p* < 0.05.

The calculations of the correlations show trivial, statistically insignificant relationships between the postural control and proprioception variables, ranging from r = 0.01 to r = 0.09.

Weak correlations were found between the degree of repositioning error and proprioception and between the variables MV_AP_ ROE, MV_AP_ RCE and total area with regard to postural control. All the correlation matrices are shown in Table 2.

A mediation analysis was carried out to assess the potential mediating role of the variable measuring the variations in postural control with open and closed eyes (VAR_CE) in the relationship between proprioception and the variables reflecting postural control with open eyes. The results are shown in Table 3. No significant total or direct effect was found for any of the variables. However, a significant indirect effect was found for the variables MV_AP_ RCE and MV_ML_ RCE, but not for total area. In other words, these results show a total mediation of MV_AP_ RCE and MV_ML_ RCE in the relationship between proprioception and postural control, but no mediating effect when it comes to the variable of total area.

## 4. Discussion

The aim of this study was to analyze the mediating role of vision in the relationship between conscious lower limb proprioception (knee of the dominant leg) and bipedal postural control (with open and closed eyes) in older adults, as compared with teenagers, younger adults and middle-aged adults. Overall, we found that vision did not play a mediating role between conscious proprioception and postural control. However, the absence of vision did have an indirect effect on the relationship between proprioception and balance. Specifically, while the older adults performed similarly well on the balance test as middle-aged and younger adults, they did slightly better than the teenagers on this test. Meanwhile, all the age groups performed worse on the balance test with their eyes closed, with the exception of the older adults, for whom no performance differences were found between the open and closed eyes tests for some of the variables. As age increases, no performance differences were found between the groups on the proprioception test.

The mediation analysis also showed a significant indirect effect of proprioception on the postural control variables MV_AP_ and MV_ML_ with closed eyes, but not with total area. To be more specific, on the bipedal postural control test with their eyes closed, participants in all the age groups (except the older adults) scored higher for the displacement of the center of gravity. In other words, in the absence of vision, for all the age groups with the exception of the oldest adults, proprioception serves as the main source of stimuli necessary to inform postural control in a specific joint [13]. However, when visual stimuli are available, the thalamus gives greater priority to this information and transmits it to the cerebral cortex, meaning that visual data can stand in the way of greater knowledge of joint movement and positioning [35]. However, the lack of difference between the postural control scores recorded by the older adults with their eyes open and with their eyes closed suggests that, for these older people, proprioception is in both cases the main source of information. We might also conclude that older operate differently from the other age groups when it comes to integrating the available information in the brain’s central processing centers. These results echo findings by [16], who tested 95 older adults and found that vision is not a mediating factor in postural balance but that proprioceptive information does play an important role in postural control. Another study in the same vein by [17] also found that the postural control of older people was not affected when visual information was compromised. The changes in the strategies we use to integrate postural control information as we get older might be due to an overall deterioration of the central nervous system that affects our proprioceptive acuity, as suggested by [17]. Two other studies using functional magnetic resonance imaging [36,37] found that an extended network of neuronal activation tends to appear in older people when their balance is tested (especially among those who perform the best). This activation includes a secondary somatosensory area, as well as other areas involved in sensorial integration such as the superior temporal gyrus and the supramarginal gyrus. This increase in brain activation might be part of a strategy to compensate for age-related decline in sensory-motor and cognitive processing.

Although both postural control and proprioception involve the use of mechanoreceptors, the sensory information the receptors provide is processed in different ways. The sensory-motor system is mediated by the signals that allow for the integration of sensorial information, which it uses to control and carry out movements. This information comes from proprioceptive receptors and other mechanoreceptors such as vestibular and tactile systems [6,38,39]. However, the stimuli that come from these proprioceptive, vestibular and tactile mechanoreceptors is processed in a range of different areas and locations within the primary sensorial cortex [40]. This could be one explanation for the lack of an association between the stability data gathered in the postural control tests and the scores for joint repositioning, suggesting that proprioceptive receptors are not necessarily activated during postural control exercises [22]. In contrast, though, other researchers examining the links between postural control and proprioception with different joints have found some correlations. One study, for example, found a link between proprioception of the knee at a 45° angle and both static and dynamic postural control. The authors explain this correlation by saying that when the knee is at a 45º angle, the quadriceps exerts more force, thus giving greater support to the bent knee and reinforcing the stability of both the knee and the center of gravity, which is related to postural control [41]. Despite these findings, the authors of these studies have observed that there is no single method that can be used on its own to accurately assess an individual’s to feel and balance him- or herself [22,41].

On the postural control tests, the group of older adults outperformed only the group of teenagers. This poor performance by the teenagers in the study might be due to the late maturation of the nervous system, as it seems that teenagers are unable to use proprioceptive information for postural control. This ability might only develop fully in adulthood, meaning that teenagers would use different postural strategies than those employed by adults [42]. In contrast with this study, other authors have reported worse performance on balance tests by older adults than by younger and middle-aged adults. These discrepancies in the findings of the different studies might be due to differences in sample size (usually *n* > 15 participants) [10,26]; or to subjecting participants to balance tests of varying levels of difficulty (for example, unilateral tests) [26]. Alternatively, the differences might be explained by differing physical fitness levels of the older participants, as physical exercise seems to be a protective factor against age-related cognitive decline [43]. The older people recruited to participate in our study were very physically active and did at least an hour a day of physical exercise, a factor that has been linked to a lesser likelihood of balance problems [44]. Furthermore, teenagers stated less physical activity than older adults across the week (only twice a week), which could explain the differences between these groups. Additionally, the participants in the sample here did the recommended amount of physical activity, which is rare among older adults [45]. Nevertheless, further research is needed to add explanation for differences between older adults and teenagers. A recent review and metanalysis by [46] that total balancing area can distinguish people who tend to fall from those who do not. The fact that there was no difference between the performance on the balance test of the older adults in our sample and that of the younger and middle-aged adults underlines the good physical fitness of the older participants.

Meanwhile, no differences were found between the older adults and the rest of the participants in conscious proprioception. This lack of difference in proprioception scores might be attributable to the fact that the older adults in our sample were physically active and had conserved their proprioceptive abilities to a very satisfactory degree, as sensory-motor deficits tend to be associated with the aging process [8] and physical activity seems to be a protective factor against age-related deterioration [43].

For example, older adults of the same age (>o = 70 years) can display different degrees of deterioration. Those with poor lower-limb proprioception tend to have worse balance than those with very good proprioception [47]. It is worth highlighting, then, that unlike in other studies, the sample of older adults here might have very good proprioception because of their active lifestyle.

The fact that this study did not find a correlation between postural control and proprioception does not mean that these two perceptual-motor skills are entirely independent of one another. Many well-established training methods use balancing tasks to improve proprioception despite the evidence that proprioceptive receptors are not activated during postural control work [48]. However, proprioception does play a fundamental role in postural control, as the information gathered by the proprioceptive system allows one to make adjustments in the control and execution of movements through the cerebellum, the cerebral cortex and the thalamus [40], thus reducing the risk of injury during exercise and everyday life [49,50]. Therefore, proprioception exercises can make a positive contribution to the functioning of dynamic and static postural control [51]. Although some studies have shed light on the link between proprioception and postural control in people with ACL injuries [22], there is a dearth of evidence on how these links manifest themselves in healthy individuals, and there is even less data on older people. The results presented here suggest that while proprioceptive information may play an important role in helping older adults maintain postural control, the performance of the participants on the proprioception tests displayed only an indirect correlation with their scores for postural control with their eyes closed.

The measurement in this study focused on the knee, because, along with the hips, the toes and the ankles, this joint has the primary responsibility for providing the body with balance. In fact, the knee ligaments offer a degree of stabilization that is fundamental for backward and forward movements as well as for external and internal rotation [20]. At the same time, the placement of the knee at a 45° angle prompts changes in the hemodynamics of the knee and ankle, and, in the sports world, this is a fundamental concern of lower limb injury prevention and rehabilitation strategies [52]. For this reason, in order to eliminate the effects of an initial adaptation by the toes and ankles and to focus the proprioception and postural control tests solely on the knees (controlled by the stability offered by the hips), the participants were outfitted with orthopedic boots. While previous studies have used ski boots [53,54], the orthopedic boots used here allowed us to collect more accurate measurements of the knees, as they allowed the participants to bend their knees freely without any accompanying dorsiflexion of the foot. Additionally, we ensured that the participants did not make any hip movements to balance themselves with the help of a member of the research team who confirmed that each participant’s torso remained vertical at all times, thus further guaranteeing the focus on postural control mechanisms in the knee. It is worth noting that this is not the first study focusing on the knee that has corrected for adaptations by the toes and ankles (which also play key roles in postural control) [22], but, to our knowledge, it is the first to have done so using orthopedic boots.

Finally, some earlier research has concluded that a combination of regular proprioception and postural control exercises can be a beneficial way to prevent knee injuries among all age groups [21], possibly even offering additional benefits to older people in that it helps guard against falls [55]. Additionally, training programs featuring proprioception exercises can also be effective at improving motor functioning [56,57], as some authors suggest that physical exercise can counteract the effects of aging on proprioceptive ability [7]. Meanwhile, correlations have been found between performance on proprioception tests and the number of falls experienced over the previous 12 months in a group of 95 older adults (Lord et al. 1991). Among older adults, conscious proprioception of the knee specifically has been linked to a greater volume of oxygen, although no causal relations have been established [58]. Nonetheless, there is evidence that with training older people can improve both their proprioception and their balance. For example [58], showed that five weeks of proprioception training improved the motor control of a group of 20 older people with respect to a group that did not receive the training. It is worth noting, though, that improvements in proprioception and postural control among older people seem to be limited to the specific training regimen applied [7].

In light of the results here, trainers should be aware of the role of vision. They should also take into account that exercises aimed at improving the conscious control of the positioning of a given body part should be done with the eyes closed. Meanwhile, if the objective is to improve overall postural control, exercises can be done with the eyes open so as to facilitate the participation of visual, proprioceptive and vestibular information in the carrying out of the task.

## 5. Conclusions

To our knowledge, this is the first study to show an indirect mediation between proprioception and postural control with the eyes closed. Only trivial and statistically insignificant correlations were found between proprioception and bipedal postural control overall. Specifically, older adults outperformed only teenagers on the balance test. At the same time, the group of older adults was the only one that did not display differences with regard to certain variables when the test was done with open or closed eyes. Additionally, it seems that age does not influence performance on proprioception tests. These findings are important in they suggest that functional rehabilitation and readaptation programs for older people should include tasks without vision in order to improve conscious control of the positioning of body parts. This, in turn, will help improve postural control and minimize the risk of falls.

## Figures and Tables

**Figure 1 healthcare-10-00103-f001:**
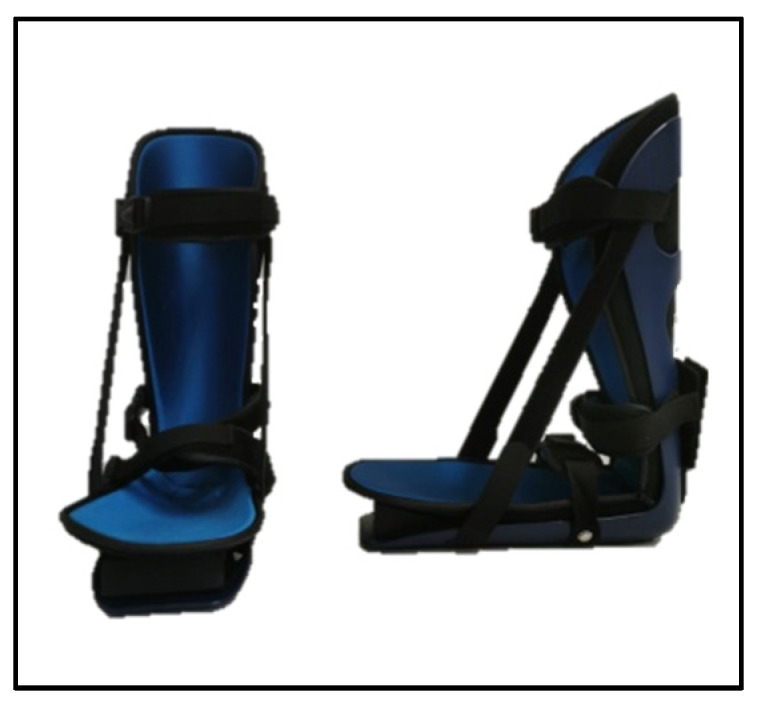
The orthopedic boots used in the study.

**Figure 2 healthcare-10-00103-f002:**
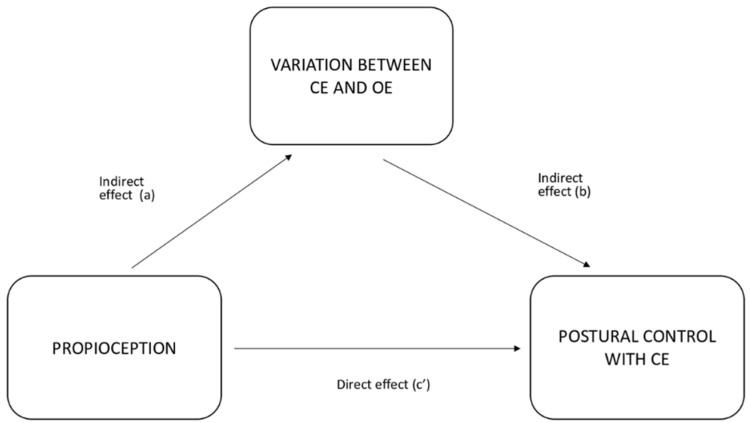
Mediation analysis associations.

**Figure 3 healthcare-10-00103-f003:**
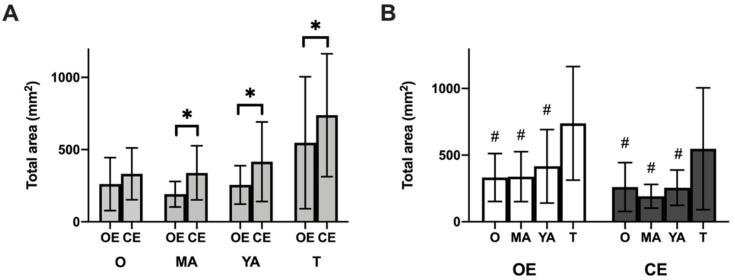
Scores on the balance test for the variable total area of center or pressure. (**A**) Comparison by testing conditions (open eyes vs. closed eyes) and (**B**) Comparison by age group. Lower values are associated with better performance on the task. O: Older adults; MA: middle-aged adults YA: young adults; T: teenagers; OE open eyes; CE: closed eyes. * *p* < 0.001. # post hoc differences only vs. the group of teenagers, *p* < 0.001.

**Figure 4 healthcare-10-00103-f004:**
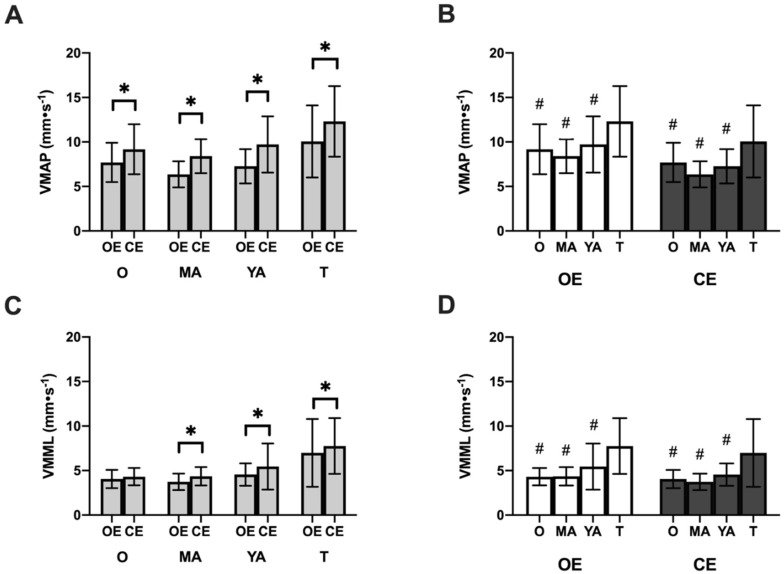
Scores on the balance test for the variable mean velocity of center of pressure. Mean front-to-back velocity: (**A**) Comparison by conditions (open eyes vs. closed eyes) and (**B**) Comparison by age groups. Mean velocity in medial-lateral direction: (**C**) Comparison by conditions (open eyes vs. closed eyes) and (**D**) Comparison by age group. Lower scores indicate better performance on the task. O: older adults; MA: middle-aged adults; YA: young adults; T: teenagers; OE: open eyes; CE: closed eyes; MV_AP_: media front to back velocity of the center of pressure; MV_ML_: median medial lateral velocity of the center of presser. * *p* < 0.001. # post hoc differences only vs. the group of teenagers, *p* < 0.001.

**Figure 5 healthcare-10-00103-f005:**
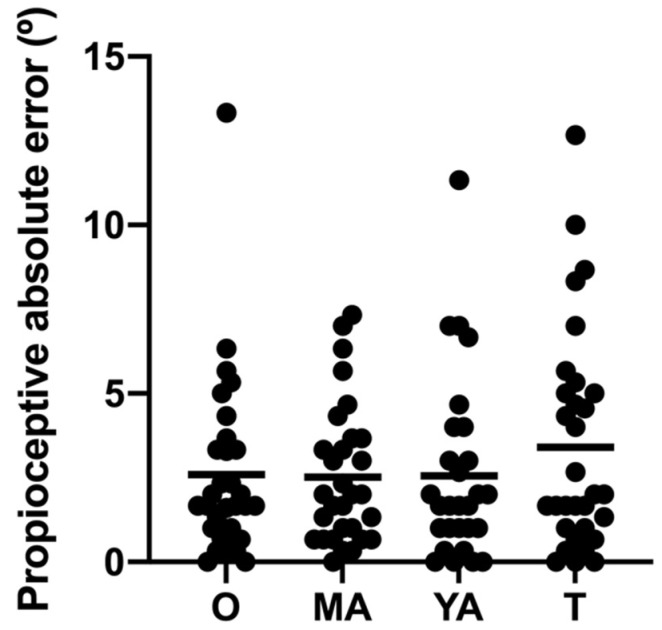
Performance on the proprioception test.

**Table 1 healthcare-10-00103-t001:** Demographic characteristics of the participants.

Group	*n*	Age (Years)	Weight (Kg)	Body Mass Index (Kg/m^2^)	Physical Activity Level (METS)
Teenagers	30	14.4 ± 1.7	58.1 ± 11.1	21.0 ± 3.0	3320.9 ± 1204.6
Young adults	29	23.6 ± 2.7	62.2 ± 9.7	22.1 ± 2.2	4219.0 ± 1824.2
Middle-aged adults	30	45.4 ± 6.7	68.5 ± 24.6	23.9 ± 3.7	6976.3 ± 2898.0
Older adults	30	73.5 ± 5.9	65.8 ± 7.1	25.8 ± 2.4	3927.8 ± 1569.3

Data are expressed as means ± standard deviation. Note: data on the participants’ physical activity level were gathered using the validated IPAQ questionnaire.

**Table 2 healthcare-10-00103-t002:** Correlation coefficients between proprioception and postural control with eyes open and closed.

	MV_AP_ ROE	MV_ML_ ROE	Área Total ROE	MV_AP_ RCE	MV_ML_ RCE	Total Area RCE
MV_ML_ ROE	0.79 *	—				
Area total ROE	0.74 *	0.80 *	—			
MV_AP_ RCE	0.78 *	0.63 *	0.55 *	—		
MV_ML_ RCE	0.70 *	0.81 *	0.63 *	0.79 *	—	
Total Area RCE	0.65 *	0.67 *	0.72 *	0.72 *	0.82 *	—
Propioception	−0.06	0.01	0.05	0.09	0.09	0.03

ROE = Romberg open eyes; RCE = Romberg closed eyes; MV_AP_ = median front-to-back velocity: MV_ML_ = median medial-lateral velocity of center of pressure. * *p* < 0.05

**Table 3 healthcare-10-00103-t003:** Approximate mediating role of the variable (VAR_CE) in the relationship between proprioception and the variables measuring postural control with eyes closed.

Postural Control Variable	Effect	Coefficient	Confidence Interval	t	*p*-Value	% Mediation
MV_AP_	Indirect (a × b)	0.070	0.008; 0.132	2.223	0.026 *	88.1
	Direct (c)	0.009	−0.147; 0.167	0.118	0.906	11.9
	Total (c + a × b)	0.079	−0.078; 0.238	0.988	0.323	100.0
MV_ML_	Indirect (a × b)	0.045	0.002; 0.089	2.050	0.040 *	74.1
	Direct (c)	0.016	−0.103; 0.135	0.263	0.792	25.9
	Total (c + a × b)	0.061	−0.058; 0.182	1.009	0.313	100.0
Total Area	Indirect (a × b)	−2.12	−6.73; 2.49	−0.902	0.367	34.5
	Direct (c)	4.04	−9.19; 17.26	0.598	0.550	65.5
	Total (c + a × b)	1.92	−12.00; 15.83	0.270	0.787	100.0

* *p* < 0.05 indicates statistically significant differences.

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
