# Peer review of "The Mediating Role of Vision in the Relationship between Proprioception and Postural Control in Older Adults, as Compared to Teenagers and Younger and Middle-Aged Adults"

_healthcare, 2022, doi:10.3390/healthcare10010103_

Round 1

Reviewer 1 Report

I consider it to be a very interesting study that investigates the relationship and mediating role of vision with conscious proprioception in different age groups.

I see the methodological design as totally adequate, as well as the type of statistical analysis used.

The writing of the text is very careful and there are no relevant syntactic errors.

I liked the discussion very much and I see it as very well founded, I am thankful for it.

I suppose that the text in its current state, will have already passed some kind of prior review.However, some small changes are suggested that could help to make it perfect.

I think the abstract should be reformulated. The objectives described in it (design of training programs for older adults) do not coincide with those described in the last paragraph of the introduction, nor do the conclusions of the study.

Acronym revision is necessary. there are some whose meaning is not described in the text. For example IPAQ (International Physical Activity Questionnaire) or METs (metabolic equivalents)

It would be interesting to incorporate some type of code linked to the approval of the deontological committee of the university.

Author Response

Editor comments to Author:

Responses to Editor

Please note that:

[E] = comments from Editor

[R1] = comments from Reviewer #1.

[R2] = comments from Reviewer #2.

[R3]= comments from Reviewer #2

[A] = answers from the authors.

{…} = text modified in the revised manuscript.

Reviewer 1:

[R1] =

  1. I think the abstract should be reformulated. The objectives described in it (design of training programs for older adults) do not coincide with those described in the last paragraph of the introduction, nor do the conclusions of the study.
  2. Acronym revision is necessary. there are some whose meaning is not described in the text. For example IPAQ (International Physical Activity Questionnaire) or METs (metabolic equivalents)
  3. It would be interesting to incorporate some type of code linked to the approval of the deontological committee of the university.

[A] = We have amended as follow:

  1. Line 21-32 {…Abstract: The aim of this study is to analyze the mediating role of vision in the relationship between conscious lower limb proprioception (dominant knee) and bipedal postural control (with eyes open and closed) in older adults, as compared with teenagers, younger adults and middle-aged adults. Methods: The sample consisted of 119 healthy, physically active participants. Postural control was assessed using the bipedal Romberg test with participants’ eyes open and closed on a force platform. Proprioception was measured through the ability to reposition the knee at 45°, measured with the Goniometer Pro application's goniometer. Results: The results showed an indirect relationship between proprioception and postural control with closed eyes in all age groups; however, vision did not mediate this relationship. Conclusions: Older adults outperformed only teenagers on the balance test. The group of older adults was the only one that did not display differences with regard to certain variables when the test was done with open or closed eyes. It seems that age does not influence performance on proprioception tests. These findings help us to optimize the design of training programs for older adults and suggest that physical exercise is a protective factor against age-related decline. }
  2. Line 136-137:

{…Participants: The overall sample for this study was made up of 119 physically active individuals [according to the results of the International Physical Activity Questionnaire, (IPAQ) 4611.03±1620.97 Metabolic Equivalents (METS)] of between 12 and 85 years of age.}

  1. Line 162-165:

{…The study was carried out in accordance with the ethical standards set out in the Helsinki Declaration, and it was approved by the ethics committee of Universitat Ramon Llull, ID number 1718006D. All the participants in the study were informed of the procedure and signed informed consent documents.}

Reviewer 2 Report

Why was the gender variable not considered in the study?

In the group of older adults there is a BMI that reflects an overweight, unlike the other groups where the BMI falls within normality, do not you think that this fact has been able to influence the measurement of postural control?

It is surprising that the group of older adults (65-85) has a higher physical activity level than the group of teenegers (12-18). What do you think it may be due to?

The text says: "A few days before carrying out the measurement protocol, the researchers visited the institutions from which the participants had been recruited, in order to collect data on their medical records." I would like this to be clarified because medical records are confidential and private and an investigator cannot go to the hospital to ask for these data.

The Romberg test is normally performed barefoot. When including orthopedic boots in your realization, it would be convenient to specify that it is a modified Romberg.

Author Response

Editor comments to Author:

Responses to Editor

Please note that:

[E] = comments from Editor

[R1] = comments from Reviewer #1.

[R2] = comments from Reviewer #2.

[R3]= comments from Reviewer #2

[A] = answers from the authors.

{…} = text modified in the revised manuscript.

Reviewer 2:

[R2] =

  1. Why was the gender variable not considered in the study?
  2. In the group of older adults there is a BMI that reflects an overweight, unlike the other groups where the BMI falls within normality, do not you think that this fact has been able to influence the measurement of postural control?
  3. It is surprising that the group of older adults (65-85) has a higher physical activity level than the group of teenegers (12-18). What do you think it may be due to?
  4. The text says: "A few days before carrying out the measurement protocol, the researchers visited the institutions from which the participants had been recruited, in order to collect data on their medical records." I would like this to be clarified because medical records are confidential and private and an investigator cannot go to the hospital to ask for these data.
  5. The Romberg test is normally performed barefoot. When including orthopedic boots in your realization, it would be convenient to specify that it is a modified Romberg.

[A] = We have amended as follow:

  1. The gender variable not considered in the study because the sample was not homogeneous and by gender and the main purpose was to analyze differences by age groups.
  2. We think that is unlikely that BMI could influence the measurement of postural control, because although older adults participants BMI was 25.8 ±2.4 their mean weight (65.8 ±7.1Kg) was lower than Middle-aged adults (68.5 ±24.6Kg) and this weight would be a fact that could influence the measurement of postural control on the force platform.
  3. The older people recruited to participate in our study were very physically active and did at least an hour a day of physical exercise, a factor that has been linked to a lesser likelihood of balance problems. Additionally, the participants in the sample here did the recommended amount of physical activity, which is rare among older adults. In contrast, teenagers participant only state to practice their sports twice per week and once Physical Education session at the school.
  4. Line 168-173 {…The researches collected data by an anonymous formular sent by email to the participant. Prior sending the email we check that older adults were familiar with internet and we gave them verbal instructions to fulfilling the formular. The formular included information on the participants’ physical characteristics, along with data to determine whether the individuals fulfilled the inclusion/exclusion criteria, not medical history was collected.}

5. Although we use the orthopedic boots in our realization there was no change in Romberg protocol. Therefore we considered that is was not a modified Romberg test. Other researchers considered a modified Romberg test by adding a foam under the barefoot because it could reduce the applied force on the platform. Nevertheless, we didn’t observe any reduction in the detected pressure with the orthopedic boots, and this is the reason why we decided not to call it “modified Romberg”.

Reviewer 3 Report

Introduction: The Introduction was clearly presented. However, I would add how balance works and describe the integration of multisensory processes centrally to better explain to the reader why it is important to investigate the mediating role of vision in the relationship between conscious knee proprioception.
Methodology: I am sorry If I have missed it but I would try to add how you controlled for learning patterns during the 3 tests you measured. 
Results: the results were clearly exposed.
Discussion: The discussion was well presented, however, the argument remains weak on why the elderly out-performed the younger. If you have any other hypotheses or future assessments you would do to confirm the results please mention them in the discussion. 
Would you perform other studies to assess how visual inputs are integrated during this measurement? for example: provocative visual stimuli, to assess whether the elderly group is able to better suppress visual inputs and to rely more on proprioception control?or not? 

Author Response

Editor comments to Author:

Responses to Editor

Please note that:

[E] = comments from Editor

[R1] = comments from Reviewer #1.

[R2] = comments from Reviewer #2.

[R3]= comments from Reviewer #2

[A] = answers from the authors.

{…} = text modified in the revised manuscript.

Reviewer 3:

[R3] =

1.Introduction: The Introduction was clearly presented. However, I would add how balance works and describe the integration of multisensory processes centrally to better explain to the reader why it is important to investigate the mediating role of vision in the relationship between conscious knee proprioception.

  1. Methodology: I am sorry If I have missed it but I would try to add how you controlled for learning patterns during the 3 tests you measured. 

  2. Results: the results were clearly exposed.

  3. Discussion: The discussion was well presented, however, the argument remains weak on why the elderly out-performed the younger. If you have any other hypotheses or future assessments you would do to confirm the results please mention them in the discussion. 

  4. Would you perform other studies to assess how visual inputs are integrated during this measurement? for example: provocative visual stimuli, to assess whether the elderly group is able to better suppress visual inputs and to rely more on proprioception control?or not? 

[A] = We have amended as follow:

  1. Line 41-63 {.. This motor skill is closely connected to the central nervous system, and it evolves with age (Park et al., 2016), reaching complete maturity in adulthood (> 18 years of age). This maturity is followed by a decline, when people gradually lose synapses and their non-functional neural connections disappear, causing a deterioration of motor skills among older adults (Pérez et al., 2017). One way this manifests itself is that older adults tend to display greater variability in the center of pressure than younger adults when carrying out postural tasks (Delmas et al., 2021). Postural control depends on the interaction of proprioceptive receptors, the sense of touch, the vestibular system and the processing of visual information, all of which helps regulate tone and the perception of force and pressure (Gaerlan, 2010). The degree to which this skill reaches its highest potential depends on the integration of visual, vestibular, proprioceptive and tactile inputs, which play a role in regulating the tonicity and the perception of strength and pressure needed to keep the balance (Gaerlan 2010).}
  2. Line 244-248 :

{…. The order in which the tests were administered was chosen randomly for each participant, and the tests were repeated three times under each set of testing circumstances, with each test taking thirty seconds. Learning patterns were avoided by the researchers by keeping in silence and not giving any feedbacks across the three attempts.}

  1. Thank you very much for your comment.
  2. Line 426-434 {… The older people recruited to participate in our study were very physically active and did at least an hour a day of physical exercise, a factor that has been linked to a lesser likelihood of balance problems (Gomez-Bruton et al., 2020). Furthermore, teenagers stated less physical activity than older adults across the week (only twice a week), which could explain the differences between these groups. Additionally, the participants in the sample here did the recommended amount of physical activity, which is rare among older adults (Sun et al., 2013). Nevertheless, further research is needed to add explanation for differences between older adults and teenagers. …}
  3. Yes, it would be very interesting to perform other studies to assess how visual inputs are integrated during this measurement. We will consider your suggestions for future research. Thank you very much.
